# FEVEROUS: Fact Extraction and VERification Over Unstructured and Structured information

**Rami Aly, Zhijiang Guo, Michael Schlichtkrull, James Thorne, Andreas Vlachos**
Department of Computer Science and Technology
University of Cambridge
{rmya2,zg283,mss84,jt719,av308}@cam.ac.uk

**Christos Christodoulopoulos**
Amazon Alexa
chrchrs@amazon.co.uk

**Oana Cocarascu**
Department of Informatics
King's College London
oana.cocarascu@kcl.ac.uk

**Arpit Mittal**
Facebook*
arpitmittal@fb.com

## Abstract

Fact verification has attracted a lot of attention in the machine learning and natural language processing communities, as it is one of the key methods for detecting misinformation. Existing large-scale benchmarks for this task have focused mostly on textual sources, i.e. unstructured information, and thus ignored the wealth of information available in structured formats, such as tables. In this paper we introduce a novel dataset and benchmark, Fact Extraction and VERification Over Unstructured and Structured information (FEVEROUS), which consists of 87,026 verified claims. Each claim is annotated with evidence in the form of sentences and/or cells from tables in Wikipedia, as well as a label indicating whether this evidence supports, refutes, or does not provide enough information to reach a verdict. Furthermore, we detail our efforts to track and minimize the biases present in the dataset and could be exploited by models, e.g. being able to predict the label without using evidence. Finally, we develop a baseline for verifying claims against text and tables which predicts both the correct evidence and verdict for 18% of the claims.

## 1 Introduction

Interest in automating fact verification has been growing as the volume of potentially misleading and false claims rises [Graves, 2018], resulting in the development of both fully automated methods (see Thorne and Vlachos [2018], Zubiaga et al. [2018], Hardalov et al. [2021] for recent surveys) as well as technologies that can assist human journalists [Nakov et al., 2021]. This has been enabled by the creation of datasets of appropriate scale, quality, and complexity in order to develop and evaluate models for fact extraction and verification, e.g. Thorne et al. [2018], Augenstein et al. [2019]. Most large-scale datasets focus exclusively on verification against textual evidence rather than tables. Furthermore, table-based datasets, e.g. Chen et al. [2020a], assume an unrealistic setting where an evidence table is provided, requiring extensions to evaluate retrieval [Schlichtkrull et al., 2020].

---

*The author started working on this project whilst at Amazon.

35th Conference on Neural Information Processing Systems (NeurIPS 2021) Track on Datasets and Benchmarks.

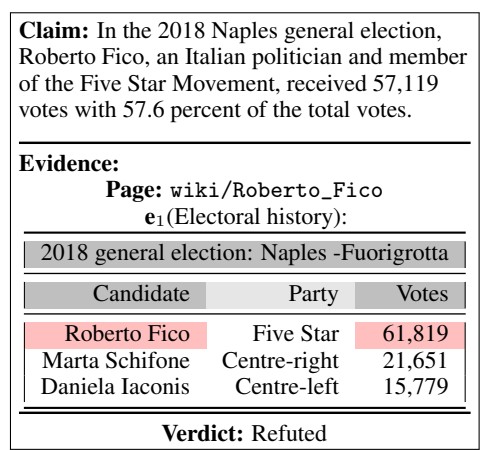

Figure 1: FEVEROUS sample instances. Evidence in tables is highlighted in red. Each piece of evidence $e_i$ has associated context, i.e. page, section title(s) and the closest row/column headers (highlighted in dark gray). Left: evidence consists of two table cells refuting the claim. Right: Evidence consists of two table cells and one sentence from two different pages, supporting the claim.

In this paper, we introduce a novel dataset and benchmark, FEVEROUS: Fact Extraction and VERification Over Unstructured and Structured information, consisting of claims verified against Wikipedia pages and labeled as supported, refuted, or not enough information. Each claim has evidence in the form of sentences and/or cells from tables in Wikipedia. Figure 1 shows two examples that illustrate the level of complexity of the dataset. A claim may require a single table cell, a single sentence, or a combination of multiple sentences and cells from different articles as evidence for verification. FEVEROUS contains 87,026 claims, manually constructed and verified by trained annotators. Throughout the annotation process, we kept track of the two- and three-way inter-annotator agreement (IAA) on random samples with the IAA kappa $\kappa$ being 0.65 for both. Furthermore, we checked against dataset annotation biases, such as words present in the claims that indicate the label irrespective of evidence [Schuster et al., 2019], and ensured these are minimised.

We also develop a baseline approach to assess the feasibility of the task defined by FEVEROUS, shown in Figure 2. We employ a combination of entity matching and TF-IDF to extract the most relevant sentences and tables to retrieve the evidence, followed by a cell extraction model that returns relevant cells from tables by linearizing them and treating the extraction as a sequence labelling task. A RoBERTa classifier pre-trained on multiple NLI datasets predicts the veracity of the claim using the retrieved evidence and its context. This baseline substantially outperforms the sentence-only and table-only baselines. The proposed baseline predicts correctly both the evidence and the verdict label for 18% of the claims. The retrieval module itself fully covers 28% of a claims evidence. FEVEROUS is the first large-scale verification dataset that focuses on sentences, tables, and the combination of the two, and we hope it will stimulate further progress in fact extraction and verification and is publicly available online: `https://fever.ai/dataset/feverous.html`.

## 2 Literature Review

Datasets for fact verification often rely on real-world claims from fact-checking websites such as PolitiFact. For such claims, the cost of constructing fine-grained evidence sets can be prohibitive. Datasets therefore either leave out evidence and justifications entirely [Wang, 2017], rely on search engines which risk including irrelevant or misleading evidence [Popat et al., 2016, Baly et al., 2018, Augenstein et al., 2019], or bypass the retrieval challenge entirely by extracting evidence directly from the fact checking articles [Alhindi et al., 2018, Hanselowski et al., 2019, Kotonya and Toni, 2020b] or scientific literature [Wadden et al., 2020].

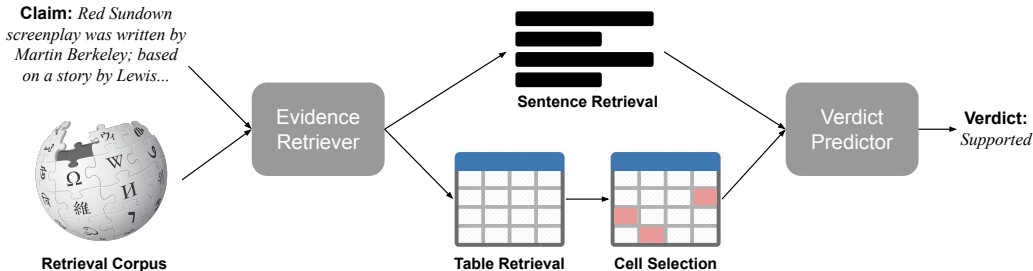

Figure 2: The pipeline of our FEVEROUS baseline.

The cost of curating evidence sets for real-world claims can be circumvented by creating artificial claims. Thorne et al. [2018] introduced FEVER, a large-scale dataset of 185,445 claims constructed by annotators based on Wikipedia articles. This annotation strategy was adopted to construct a similar dataset for Danish [Nørregaard and Derczynski, 2021], and adapted for real-world climate change-related claims [Diggelmann et al., 2021]. Jiang et al. [2020] extended this methodology to create a dataset of 26k claims requiring multi-hop reasoning. Other annotation strategies include Khouja [2020] who introduced a dataset of Arabic claims generating supported and unsupported claims based on news articles.

The datasets discussed so far have primarily focused on unstructured text as evidence during the annotation process. There is currently a small number of datasets that rely on structured information, primarily tables. TabFact [Chen et al., 2020a] and InfoTABS [Gupta et al., 2020] contain artificial claims to be verified on the basis of Wikipedia tables and infoboxes respectively, while SEM-TAB-FACTS [Wang et al., 2021] requires verification on the basis of tables from scientific articles. The latter is the only to also specify the location of the evidence in a table. Our proposed dataset is the first which considers both structured *and* unstructured evidence for verification, while explicitly requiring the retrieval of evidence.

In the related field of question answering [Bouziane et al., 2015], recent work also considered finding answers over both tables and text. Chen et al. [2020b] proposed HybridQA, a dataset consisting of multi-hop questions constructed by using Wikipedia tables and the introductory section of linked entities in the table, however, their dataset assumes the table as part of the input. Based on HybridQA, Chen et al. [2021] further required systems to retrieve relevant tables and texts by decontextualizing questions of HybridQA and adding additional questions to remove potential biases, resulting in a total of 45K question-answer pairs. The NaturalQA dataset [Kwiatkowski et al., 2019] is substantially larger (about 300K) with some questions requiring to retrieve answers from tables (17%). However, these tables are predominantly infoboxes and rarely require systems to combine information from both text and tables.

## 3 FEVEROUS Dataset and Benchmark

In FEVEROUS the goal is to determine the veracity of a claim $c$ by: *i)* retrieving a set of evidence pieces $E$ which can be either a sentence or a table cell, and *ii)* assigning a label $y \in \{\text{Supports}, \text{Refutes}, \text{Not Enough Info}\}$. The source of evidence is derived from the English Wikipedia (excluding pages and sections flagged to require addition references or citations), and consists of sentences and tables obtained as follows:

**Sentence.** Any sentence from a Wikipedia article's text as well as special Wikipedia phrases referring to other articles (e.g. *See also: ...*, *X redirects here. For other uses, see ...*).

**Table.** A table consists of cells $c_{i,j}$, where $i$ and $j$ specify the row and column, respectively, and a caption $q$. Both cells and captions can take various formats like a single word, number or symbol, phrases, and entire sentences. In most datasets (e.g. Chen et al. [2020a]), headers are restricted to the first row of a table. However, tables in Wikipedia can have a more complex structure, including multi-level headers (see Figure 1 for an example). FEVEROUS maintains the diversity of Wikipedia tables, only filtering out those with formatting errors. For the purposes of annotation, a table caption $q$ is considered to be a table cell and needs to be selected explicitly if it contains information relevant

to the claim. We also include Wikipedia infoboxes as tables, as well as lists. We consider the latter to be special tables where the number of items in the list yields the number of columns and the number of nested lists yields the number of rows. For example, $c_{1,5}$ represents the item of a nested list at depth 1 found at the fifth position of the main list.

The evidence retrieval in FEVEROUS considers the entirety of a Wikipedia article and thus the evidence can be located in any section of the article except the reference sections. The order between all elements in an article is maintained. We associate each candidate piece of evidence with its *context*, which consists of the article's title and section titles, including the sub-sections the element is located in. For table cells, we also include the nearest row and column headers; if the element just before the nearest row/column is also a header, then it will be included in the context. Context adds relevant information to understand a piece of evidence, but it is not considered a piece of evidence by itself. Sentences and cells maintain their hyperlinks to other Wikipedia articles, if present.

Quantitative characteristics of FEVEROUS and most related fact-checking datasets (i.e. FEVER, TabFact, and Sem-Tab-Facts) are shown in Table 1. As seen, the average claim of FEVEROUS is more than twice as long as the other datasets. On average $1.4$ sentences and $3.3$ cells (or $0.8$ Tables) are required as evidence per sample, higher than both FEVER and Sem-Tab-Facts *combined*. Looking into the evidence sets by type, we note that FEVEROUS is balanced, having almost an equal amount of instances containing, either exclusively text, tables, or both as evidence. Regarding the veracity labels, FEVEROUS is roughly balanced in terms of supported (56%) and refuted claims (39%), with only about 5% of claims being NotEnoughInfo.

Table 1: Quantitative characteristics of FEVEROUS compared to related datasets. Claim length is reported in tokens. *Avg. Evidence* is the average number of evidence pieces per claim in a dataset, while *Evidence Sets by type* reports the number of unique evidence sets by type. For FEVEROUS, *combined* measures the number of annotations that require evidence from both tables and sentences. The *Evidence Sets* can be used as *Evidence Source* for SEM-TAB-FACTS and TabFact, as explored by Schlichtkrull et al. [2020] for the latter.

| Statistic | FEVEROUS | FEVER | TabFact | SEM-TAB-FACTS |
|---|---|---|---|---|
| Total Claims | 87,026 | 185,445 | 117,854 | 5,715 |
| Avg. Claim Length | 25.3 | 9.4 | 13.8 | 11.4 |
| Avg. Evidence | 1.4 sentences, 3.3 cells (0.8 tables) | 1.2 sentences | 1 table | 1.1 cells (1 table) |
| Evidence Sets by Type | 34,963 sentences, 28,760 tables, 24,667 combined | 296,712 sets | 16,573 tables | 1,085 tables |
| Size of Evidence Source | 95.6M sentences, 11.8M tables | 25.1M sentences, | 16,573 tables | 1,085 tables |
| Veracity Labels | 49,115 Supported, 33,669 Refuted, 4,242 NEI | 93,367 Supported, 43,107 Refuted, 48,973 NEI | 63,723 Supported, 54,131 Refuted | 3,342 Supported, 2,149 Refuted, 224 Unknown |

## 3.1 Dataset Annotation

Each claim in the FEVEROUS dataset was constructed in two stages: *1)* claim generation based on a Wikipedia article, *2)* retrieval of evidence from Wikipedia and selection of the appropriate verdict label, i.e. claim verification. Each claim is verified by a different annotator than the one who generated it to ensure the verification is done without knowledge of the label or the evidence. A dedicated interface built on top of Wikipedia's underlying software, Mediawiki (`https://www.mediawiki.org/wiki/MediaWiki`) to give annotators a natural and intuitive environment for searching and retrieving relevant information. The ElasticSearch engine, in particular the CirrusSearch Extension, allowed for more advanced search expressions with well-defined operators and hyperlink navigation, as well as a custom built page search functionality, enabling annotators to search for specific phrases in an article. This interface allows for a diverse generation of claims, as annotators can easily combine information from multiple sections or pages. See the supplementary material for screenshots of the interface and examples of its use. We logged all of the annotators' interactions with the platform (e.g. search terms entered, hyperlinks clicked) and include them in the FEVEROUS dataset, as this information could be used to refine retrieval models with additional information on intermediate searches and search paths that led to relevant pages.

### 3.1.1 Claim Generation

To generate a claim, annotators were given a *highlight* of either four consecutive sentences or a table, located anywhere in a Wikipedia page; each page is used only once, i.e. only one set of claims is generated per page, to prevent the generation of claims that are too similar. This allowed us to control the information that annotators used and consequently the distribution of topics in the claims. Sentence highlights are restricted to sentences that have at least 5 tokens, whereas table highlights must have at least 5 rows and at most 50 rows. These bounds have been chosen based on previous work, with the rows upper bound being equal to TabFact's and the lower bounds being equal to HybridQA's. While TabFact does not use lower bounds, we noticed that it is infeasible to construct more complicated claims from tables with fewer than 5 rows.

The sentence versus table highlights ratio is 1:2. Annotators had the option to skip highlights if the sentences/tables had formatting issues or if the content enable the creation of verifiable, unambiguous, and objective claims (see supplementary material for the full list of requirements). For each highlight, annotators were asked to write three different claims with the specifications described below, each claim being a factual and well-formed sentence.

**Claim using highlight only** (Type I). This type of claim must use information exclusively from the highlighted table/sentences and their context (page/section titles or headers). For sentence highlights we did not allow claims to be paraphrases of one of the highlighted sentences, but to combine information from the four highlighted sentences instead. For claims based on a table highlight, annotators were asked to combine information from multiple cells if possible, using comparisons, filters, arithmetic and min-max operations. Only for the first claim we also specified the veracity of the generated claim, enforcing an equal balance between supported and refuted claims. This decision was motivated by the observation that annotators have a strong tendency to write supported claims as these are more natural to generate. For both Type II and III claims, annotators could freely decide to create either supported, refuted, or NEI claims, as long as they adhere to the claim requirements.

**Claim beyond the highlight** (Type II). This type of claim must be based on the highlight, but must also include information beyond it. Annotators could either modify the first claim they generated or create an unrelated claim that still included information from the highlight. Furthermore, we enforced with equal probability whether the claim had to use information exclusively from the same page or from multiple pages. For the latter, annotators were allowed to navigate Wikipedia using the search engine and page search tools previously described.

**Mutated Claim** (Type III). We asked annotators to modify one of the two claims previously generated using one of the following 'mutations': *More Specific*, *Generalization*, *Negation*, *Paraphrasing*, or *Entity Substitution*, with probabilities 0.15, 0.15, 0.3, 0.1, 0.3, respectively. These mutations are similar, but less restrictive than those used in FEVER (see supplementary material). Annotators were allowed to navigate Wikipedia freely to extract information for generating this claim.

For each generated claim, the annotators were also asked to specify the main challenge they expect a fact-checker would face when verifying that claim, selecting one out of six challenge categories: claims that require evidence from two or more sections or articles (*Multi-hop Reasoning*), combination of structured and unstructured evidence (*Combining Tables and Text*), reasoning that involves numbers or arithmetic operations (*Numerical Reasoning*), disambiguation of entities in claims (*Entity Disambiguation*), requiring search terms beyond entities mentioned in claim (*Search terms not in claim*), and *Other*.

### 3.1.2 Claim Verification

Given a claim from the previous annotation step, annotators were asked to retrieve evidence and determine whether a claim is supported or refuted by evidence found on Wikipedia. Each annotation may contain up to three possibly partially overlapping evidence sets, and each set leads to the same verdict independently. For supported claims, every piece of information has to be verified by evidence, whereas for refuted claims, the evidence only needs to be sufficient to refute one part of the claim. If the verification of a claim requires to include every entry in a table row/column (e.g. claims with universal quantification such as "highest number of gold medals out of all countries"), each cell of that row/column is highlighted. In some cases, a claim can be considered unverifiable (Not Enough Information; NEI) if not enough information can be found on Wikipedia to arrive at one of the two other verdicts. In contrast to FEVER dataset, we also require annotated evidence for NEI claims

capturing the most relevant information to verification of the claim, even if that was not possible. This ensures that all verdict labels are equally difficult to predict correctly, as they all require evidence.

Starting from the Wikipedia search page, annotators were allowed to navigate freely through Wikipedia to find relevant evidence. They were also shown the associated context of the selected evidence in order to assess whether the evidence is sufficient on its own given the context or whether additional evidence needs to be highlighted. Before submitting, annotators were shown a confirmation screen with the highlighted evidence, the context, and the selected verdict, to ensure that all required evidence has been highlighted and that they are confident in the label they have selected.

While we require information to be explicitly mentioned in the evidence in order to support a claim, we noticed that requesting the same for refuting claims would lead to counter-intuitive verdict labels. For example, "Shakira is Canadian" would be labelled as NEI when we consider the evidence "Shakira is a Colombian singer, songwriter, dancer, and record producer" *and* no mention of Shakira having a second nationality or any other relation to Canada. A NEI verdict in this case is rather forced and unnatural, as there is no reason to believe that Shakira could be Canadian given the Wikipedia article. To address these cases, we added a guideline question "Would you consider yourself misled by the claim given the evidence you found?", so that, if answered yes (as in the above example), claims are labelled as Refuted, otherwise they are labelled NEI. This label rationale is different from FEVER for which explicit evidence is required to refute a claim. While it could be argued that, our approach to labelling claims leaves potentially more room for ambiguity as the decision partially depends on what the annotator expects to find on a Wikipedia page and whether a claim adheres to the Grice's Maxim of Quantity (being as informative as possible, giving as much information as needed), our quality assessment shows that verdict agreement is very high when the annotated evidence is identical (see Section 3.2).

After finishing the verification of the given claim, annotators then had to specify the main challenge for verifying it, using the same six challenge categories as for the challenge prediction in section 3.1.1. Examples and quantitative characteristics on expected and actual challenges can be found in the supplementary material.

## 3.2 Quality Control

**Annotators:** Annotators were hired through an external contractor. A total of 57 and 54 annotators were employed for the claim generation and claim verification stages respectively.The annotations were supervised by three project managers as well as the authors of this paper. For claim generation, half of the annotators were native US-English speakers, while the other half were *language-aware* (an upper education degree in a language-related subject). English speakers from the Philippines, whereas the evidence annotators had to be language-aware native US-English speakers. The annotator candidates were screened internally by the external contractor to assess their suitability for this task. The screening followed a two-stage process. First, the candidates' English proficiency was assessed through grammatical, spelling, and fluency tests. Second, the candidates were asked to give a sentence-long summary for a given paragraph that they would then be asked to mutate by means of negation or entity substitution, similarly to Section 3.1.1. The same screening procedure was used for both tasks, with the difference that the minimum score was set higher for the claim verification part. Details on the annotator demographics can be found in the supplementary material.

**Calibration:** Due to the complexity of the annotation, we used a two-phase calibration procedure for training and selecting annotators. For this purpose, highlights with generated claims annotated with evidence and verdicts were created by the authors to cover a variety of scenarios. While the first calibration phase aimed at familiarizing the annotators with the task, the second phase contained more difficult examples and special cases. Annotators had to annotate a total of ten highlights/claims in each calibration phase. Annotations for claim generation were graded by the project managers in a binary fashion, i.e. whether a claim adheres to the guideline requirements or not and whether the expected challenge is appropriate. For claim verification they were graded using the gold annotations by the authors using label accuracy, evidence precision/recall (see Section 5.1), the number of complete evidence sets, and selected verification challenge. Before continuing with the second calibration phase, annotators had to review the scores and feedback they received in the first phase. Based on the scores in both phases, the project managers approved or rejected each annotator, with an approval

rate of $40\%$ for claim generation and $54\%$ for claim verification, with a total of $141$ and $100$ claim generation and verification candidates, respectively.

**Quality Assurance:**  Generated claims were quality checked by claim verification annotators who had to report those that did not adhere to the claim requirements, specifying the reason(s). 2534 claims were reported with the most common reason being *Ungrammatical, spelling mistakes, typographical errors*, followed by *Cannot be verified using any publicly available information*.

Around 10% of the claim verification annotations ($8474$ samples) were used for quality assurance. We measured two-way IAA using 66% of these samples, and three-way agreement with the remaining 33%. The samples were selected randomly proportionally to the number of annotations by each annotator. The $\kappa$ over the verdict label was $0.65$ both for two-way and three-way agreement. Duplicate annotations (and hence disagreements) due to measuring IAA are not considered for the dataset itself. These IAA scores are slightly lower than the ones reported for FEVER dataset ($0.68$), however the complexity of FEVEROUS is greater as entire Wikipedia pages with both text and tables are considered as evidence instead of only sentences from the introductory sections. TabFact has an agreement of $0.75$, yet in TabFact the (correct) evidence is given to the annotators. If we only look at claims where annotators chose identical evidence, verdict agreement in FEVEROUS is very high ($0.92$), showing that most disagreement is caused by the evidence annotation. Pairs of annotators annotated the same evidence for 42% of the claims and partially overlapping evidence of at least 70% for 74% of them. In 27% of the claims the evidence of one annotator is a proper subsets of another, indicating that in some cases evidence might provide more information than required, e.g. identical information that should have been assigned to two different evidence set.

Further analysing the cases of disagreement, we observe that in a third of two-way IAA disagreement cases one annotator selected NEI, which is disproportionately high considering NEI claims make up only 5% of the dataset, again indicating that the retrieval of evidence is a crucial part of the verification task. For the other two-thirds, when annotators selected opposite veracity labels we identified four sources of disagreement: (i) numerical claims that require counting a large number of cells, so small counting errors lead to opposing labels (ii) long claims with a refutable detail that had been overlooked and hence classified erroneously (iii) not finding evidence that refutes/supports a claim due to relevant pages being difficult to find (e.g. when the article's title does not appear in the claim) (iv) accidental errors/noise, likely caused by the complexity of the task. Looking into the IAA between every annotator pair shows an overall consistent annotation quality with a standard deviation of $0.07$ and a total of $10$ annotators with an average IAA of below $0.60$, and $8$ being higher than $0.70$.

**Dataset Artifacts & Biases:**  To counteract possible dataset artifacts, we measured the association between several variables, using normalized PMI throughout the annotation process. We found that no strong co-occurrence was measured between the verdict and the words in the claim, indicating that no claim-only bias [Schuster et al., 2019] is present in the dataset. We observed the following correlations: an evidence table/sentence being the first element on a page with supported verdict (nPMI=0.14) and after position 20 with NEI verdict (nPMI=0.09); words 'which/who' with Claim Type II as well as mutation type *More specific* and *Entity Substitution* (nPMI=0.07); Claim Type II with supported verdict (nPMI=0.17) and Claim Type III with refuted label (nPMI=0.23). The latter can most likely be attributed to the *Negation* and *Entity substitution* mutations. Since we do not release the claim-type correspondence, the association of words with claim types and mutations is not of concern.

We also developed a **claim-only baseline**, which uses the claim as input and predicts the verdict label. We opted to fine-tune a pre-trained BERT model [Devlin et al., 2019] with a linear layer on top and measured its accuracy using 5-fold cross-validation. This claim-only baseline achieves $0.58$ label accuracy, compared to the majority baseline being $0.56$. Compared to FEVER where a similar claim-only baseline achieves a score of about $0.62$ over a majority baseline of $0.33$ [Schuster et al., 2019], the artefacts in FEVEROUS appear to be minimal in this respect. Regarding the position of the evidence, we observed that cell evidence tends to be located in the first half of a table. For smaller tables, evidence is more evenly distributed across rows. Moreover, a substantial amount of claims require using entire columns as evidence, and thus the later parts of a table as well.

Finally, we trained a **claim-only evidence type model** to predict whether a claim requires as evidence sentences, cells, or a combination of both evidence types. The model and experimental setup were identical to the one used to assess claim-only bias. The model achieved $0.62$ accuracy, compared to

$0.43$ using the majority baseline, suggesting that the claims are to some extent indicative of the type, but a strong system would need to look at the evidence as well.

## 4  Baseline Model

**Retriever**   Our baseline retriever module is a combination of entity matching and TF-IDF using DrQA [Chen et al., 2017]. Combining both has previously been shown to work well, particularly for retrieving tables [Schlichtkrull et al., 2020]. We first extract the top $k$ pages by matching extracted entities from the claim with Wikipedia articles. If less than $k$ pages have been identified this way, the remaining pages are selected by Tf-IDF matching between the introductory sentence of an article and the claim. The top $l$ sentences and $q$ tables of the selected pages are then scored separately using TF-IDF. We set $k = 5$, $l = 5$ and $q = 3$.

For each of the $q$ retrieved tables, we retrieve the most relevant cells by linearizing the table and treating the retrieval of cells as a binary sequence labelling task. The underlying model is a fine-tuned RoBERTa model with the claim concatenated with the respective table as input. When fine-tuning, we deploy row sampling, similar to Oguz et al. [2020], to ensure that the tables used during training fit into the input of the model.

**Verdict prediction**   Given the retrieved evidence, we predict the verdict label using a RoBERTa encoder with a linear layer on top. Table cells are linearized to be used as a evidence, following Schlichtkrull et al. [2020] who showed that a RoBERTa based model with the right linearization performs better than models taking table structure into account. Linearization of a table's content enables cross-attention between cells and sentences by simply concatenating all evidence in the input of the model. For each piece of evidence, we concatenate its context ensuring that the page title appears only once, at the beginning of the evidence retrieved from it.

The verdict predictor is trained on labelled claims with associated cell and sentence evidence and their context. The FEVEROUS dataset is rather imbalanced regarding NEI labels (5% of claims), so we sample additional NEI instances for training by modifying annotations that contain both cell and sentence evidence by removing either a sentence or an entire table. We additionally explore the use of a RoBERTa model that has been pre-trained on various NLI datasets (SNLI [Bowman et al., 2015], MNLI [Williams et al., 2018], and an NLI-version of FEVER, proposed by Nie et al. [2020]).

## 5  Experiments

### 5.1  Dataset splits and evaluation

The dataset is split into a training, development and test split in a ratio of about $0.8, 0.1, 0.1$. We further ensured that all three claims generated from a highlight are assigned to the same split to prevent claims in the development and test splits from being too similar to the ones in training data. Quantitative characteristics are shown in Table 2. Due to the scarcity of NEI instances, we maintained a rough label balance only for the test set. In all splits, the number of evidence sets with only sentences as evidence is slightly higher than sets that contain only cell evidence or sets that require a combination of different evidence types.

Table 2: Quantitative characteristics of each split in FEVEROUS.

|  | **Train** | **Dev** | **Test** | **Total** |
|---|---|---|---|---|
| Supported | 41,835 (59%) | 3,908 (50%) | 3,372 (43%) | 49,115 (56%) |
| Refuted | 27,215 (38%) | 3,481 (44%) | 2,973 (38%) | 33,669 (39%) |
| NEI | 2,241 (3%) | 501 (6%) | 1,500 (19%) | 4,242 (5%) |
| Total | 71,291 | 7,890 | 7,845 | 87,026 |
| $E_{Sentences}$ | 31,607 (41%) | 3,745 (43%) | 3589 (42%) | 38,941 (41%) |
| $E_{Cells}$ | 25,020 (32%) | 2,738 (32%) | 2816 (33%) | 30,574 (32%) |
| $E_{Sentence+Cells}$ | 20,865 (27%) | 2,468 (25%) | 2062 (24%) | 25,395 (27%) |

The evaluation considers the correct prediction of the verdict as well as the correct retrieval of evidence. Retrieving relevant evidence is an important requirement, given that it provides a basic justification for the label, which is essential to convince the users of the capabilities of a verification system and to assess its correctness [Uscinski and Butler, 2013, Lipton, 2016, Kotonya and Toni, 2020a]. Without evidence, the ability to detect machine-generated misinformation is inherently limited [Schuster et al., 2020]. The FEVEROUS score is therefore defined for an instance as follows:

$$Score(y, \hat{y}, \mathbb{E}, \hat{E}) = \begin{cases} 1 & \exists E \in \mathbb{E} : E \subseteq \hat{E} \wedge \hat{y} = y, \\ 0 & \text{otherwise} \end{cases} \tag{1}$$

with $\hat{y}$ and $\hat{E}$ being the predicted label and evidence, respectively, and $\mathbb{E}$ the collection of gold evidence sets. Thus, a prediction is scored 1 iff at least one complete evidence set $E$ is a subset of $\hat{E}$ and the predicted label is correct, else 0. The rationale behind not including precision in the score is that we recognise that the evidence annotations are unlikely to be exhaustive, and measuring precision would thus penalize potentially correct evidence that was not annotated. Instead, we set an upper bound on the number of elements to be allowed in $\hat{E}$ to $s$ table cells and $l$ sentences. This distinction was made because the number of cells used as evidence is typically higher than the number of sentences. $s$ is set to 25 and $l$ to 5, ensuring that the upper bound covers the required number of evidence pieces for every instance $E$ in both development and test set.

The FEVEROUS dataset was used for the shared task of the FEVER Workshop 2021 [Aly et al., 2021], with the same splits and evaluation as presented in this paper.

## 5.2 Results

Table 3 shows the results of our full baseline compared to a sentence-only and a table-only baseline. All baselines use our TF-IDF retriever with the sentence-only and table-only baseline extracting sentences and tables only, respectively. While the sentence-only model predicts the verdict label using only extracted sentences, the the table-only baseline only extracts the cells from retrieved tables with our cell extractor model and predicts the verdict by linearising the selected cells and their context. All models use our verdict predictor for classification. Our baseline that combines both tables and sentences achieves substantially higher sores than when focusing exclusively on either sentences or tables.

Table 3: FEVEROUS scores for the sentence-only, table-only, and full baseline for both development and test set. *Evidence* measures the full coverage of evidence (i.e. Eq. 1 without the condition on correct prediction $\hat{y} = y$).

| Model | Dev | | Test | |
|---|---|---|---|---|
| | Score | Evidence | Score | Evidence |
| Sentence-only baseline | 0.13 | 0.19 | 0.12 | 0.19 |
| Table-only baseline | 0.05 | 0.07 | 0.05 | 0.07 |
| Full baseline | 0.19 | 0.29 | 0.18 | 0.29 |

**Evidence Retrieval** To measure the performance of the evidence retrieval module for retrieved evidence $\hat{E}$, we measure both the Recall@k on a document level as well as on a passage level (i.e. sentences and tables). Results are shown in Table 4. As seen for $k = 5$ the retriever achieves a document coverage of 69%. The top 5 retrieved sentences cover 53% of all sentences while the top 3 tables have a coverage of 56%, highlighting the effectiveness of our retriever to retrieve both sentences and tables. The overall passage recall is 0.55%. For comparison a TF-IDF retriever without entity matching achieves a coverage of only 49%.

Extracting evidence cells when the cell extraction model is given the gold table for the claim from the annotated data leads to a cell recall of 0.69, with a recall of 0.74 when a table contains only a single cell as evidence. Extracted cells from the retrieved tables in combination with the extracted sentences fully cover the evidence of 29% samples in the dev set.

**Verdict prediction** The right side of Table 4 shows oracle results (i.e. when given the correct evidence), as well as results without NEI sampling and without an NLI pre-trained model. Without the NEI sampling, the model is not able to recognise a single NEI sample correctly. NLI pre-training further increases results, resulting in a macro-averaged $F_1$ of 0.70.

Table 4: (left) Document and passage (sentence + tables) coverage for the retrieval module. (right) Verdict classification using gold evidence. *NLI* denotes pre-training on NLI corpora and *NEI* NEI sampling. Scores are reported in per-class $F_1$. The overall score is reported using macro-averaged $F_1$. All results are reported on the dev set.

| top | Doc (%) | Sent (%) | Tab (%) |
| --- | --- | --- | --- |
| 1 | 0.39 | 0.23 | 0.45 |
| 2 | 0.49 | 0.37 | 0.54 |
| 3 | 0.58 | 0.46 | 0.56 |
| 5 | 0.69 | 0.53 | - |

| Model | Supported | Refuted | NEI | Overall |
| --- | --- | --- | --- | --- |
| RoBERTa | 0.89 | 0.87 | 0.05 | 0.53 |
| +NLI | 0.90 | 0.88 | 0.09 | 0.62 |
| +NLI+NEI | 0.89 | 0.87 | 0.34 | 0.70 |

## 5.3 Discussion

**Retrieval of structured information.** While the verdict predictor combines information from both evidence types, our retrieval system extracts structured and unstructured information largely independently. However, tables are often specified and described by surrounding sentences. For instance [Zayats et al., 2021] enhance Wikipedia table representations by using additional context from surrounding text. Thus, sentences provide important context to tables to be understood and related to the claim (and vice versa). Moreover, we have ignored hyperlinks in our model, yet they are excellent for entity grounding and disambiguation, adding context to both tables and sentences.

**Numerical Reasoning.** An aspect currently ignored by our baseline is that a substantial number of claims in FEVEROUS require *numerical reasoning* (for about 10% of claims numerical reasoning was selected as the main verification challenge), ranging from simply counting matching cells to arithmetic operations. Dua et al. [2019] showed that reading comprehension models lack the ability to do simple arithmetic operations.

**Verification of complex claims.** Compared to previous datasets, the length of claims and number of required evidence is substantially higher. As a result, more pieces of evidence per claim need to be retrieved and related to each other. This opens opportunities to explore the effect of the order in which each part of a claim is being verified and how evidence is conditioned on each other. To facilitate research in this direction, FEVEROUS contains for each annotation a list of operations (e.g. searched ..., clicked on hyperlink ...) that an annotator used to verify a claim (see supplementary material).

**Ecological Validity** Incorporating information from both text and tables for fact-checking enables the verification of more complex claims than previous large-scale datasets, ultimately enhancing the practical relevance of automated fact-checking systems. However, FEVEROUS still simplifies real-world claims substantially, by controlling many variables of the claim generation process. For instance, it ignores the common strategy of biased evidence employed for generating misleading claims in the real world, also referred to as cherry picking, where facts which are true in isolation are being taken out of context, resulting in an overall false claim.

## 6 Conclusion

This paper introduced FEVEROUS, the first large-scale dataset and benchmark for fact verification that includes both unstructured and structured information. We described the annotation process and the steps taken to minimise biases and dataset artefacts during construction, and discussed aspects in which FEVEROUS differs from other fact-checking datasets. We proposed a baseline that retrieves sentences and table cells to predict the verdict using both types of evidence, and showed that it outperforms sentence-only and table-only baselines. With the baseline achieving a score of 0.18 we believe that FEVEROUS is a challenging yet attractive benchmark for the development of fact-checking systems.

## Acknowledgements

We would like to thank Amazon for sponsoring the dataset generation and supporting the FEVER workshop and shared task. The dataset generation and its public release was coordinated by the University of Cambridge. Rami Aly is supported by the Engineering and Physical Sciences Research Council Doctoral Training Partnership (EPSRC). James Thorne is supported by an Amazon Alexa Graduate Research Fellowship. Zhijiang Guo, Michael Schlichtkrull and Andreas Vlachos are supported by the ERC grant AVeriTeC (GA 865958).

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
