# OpenReview forum: "FEVEROUS: Fact Extraction and VERification Over Unstructured and Structured information"
_NeurIPS.cc/2021/Track/Datasets_and_Benchmarks/Round1 — NeurIPS 2021 Datasets and Benchmarks Track (Round 1)_

### Official Review · Reviewer_bRDU · 2021-07-03

**Rating:** 7
**Confidence:** 5
**Correctness:** Correct.
**Clarity:** Clear.

**Strengths:**

1. Very clear motivation to complement missing piece in FEVER community.
2. Very professional data collection pipeline.
3. Very comprehensive experiments are performed to provide enough baseline systems.

**Weaknesses:**

The paper misses a few reference papers in the QA community.

**Additional Feedback:**

no

**Documentation:**

Yes, the annotation is very comprehensive.

**Relation To Prior Work:**

Reasonable, it would be great to add these papers to reference.

1. HybridQA: A Dataset of Multi-Hop Question Answering over Tabular and Textual Data
2. Open Question Answering over Tables and Text
3. Unified Open-Domain Question Answering with Structured and Unstructured Knowledge
4. Representations for Question Answering from Documents with Tables and Text

**Summary And Contributions:**

This paper proposes a new fact verification task by combining both structured (tables) and unstructured (text) data from Wikipedia. This dataset is collected by taking into account both the crawled text and table from Wikipedia with their self-built WikiMedia interface. They use highlighting to help the annotators to annotate diverse and grounded claims. They also develop different bias mitigation strategy to alleviate potential bias introduced in the annotation procedure. They develop different baselines using text-only and table-only algorithms. The experimental results show that the combined model perform better than the individual models on the proposed dataset. However, the existing models are still obtaining very low performance due to the challenging nature of reasoning over two forms of data. The model cannot deal with numerical reasoning, does not have a great representation of the table structure, etc. These weaknesses can serve as future research directions for researchers working on this dataset. In general, I think the paper is very well written, the proposed annotation pipeline is very professional and clear. One minor concern is that the paper missed many citations of text+table QA. I think QA and FEVER are quite similar tasks, their methodologies are quite shareable. Nevertheless, I still vote to accept the paper due to its valuable contribution to the community.

---

> ### Author Response · Authors · 2021-07-12
> **Author response**
>
> We thank the reviewer very much for the feedback on our paper as well as the additional references/suggestions. We added a paragraph on related QA papers to Section 2 and added the fourth paper mentioned in the reviewer’s list as an additional reference in section 5.3.

---

> > ### Comment · Reviewer_bRDU · 2021-07-12
> > **Response**
> >
> > Thanks for adding the additional reference. I would keep my score unchanged.

---

### Official Review · Reviewer_yzsY · 2021-07-03
**A promising new dataset**

**Rating:** 8
**Confidence:** 3
**Clarity:** The writing quality is good and the p…

**Strengths:**

* Compared to the small number of datasets that tackle a similar problem, this dataset is much larger.
* The data collection process is well documented.
* The claim generation and evidence collection process seem thoughtful and reasonable.
* Tackles a timely and important problem with potentially large significant to the research community.

**Weaknesses:**

I'd have liked to see much more discussion about the ecological validity of this dataset, particularly focused on two questions:

1. How does the task design in this dataset compare to real-world fact-checking problems? To what extent could a high-performing model trained on this dataset be useful for real-world fact-checking tasks? To what extent might it be problematic?

2. The inter-rater agreement rate is quite low. The authors note that this rate is similar to that found in related datasets, and that the rate becomes very high when annotators consider the same evidence for a given claim. I'd like to have a better understanding of the  implications of this disagreements. For instance, is it often the case that different evidence results in entirely supporting or refuting the exact same claim? Or is it more often that some evidence supports a claim, while other evidence simply doesn't provide enough information? Are there clusters of annotators who tend to agree or disagree with each other, or does it appear to simply be noise?

**Additional Feedback:**

I'd like to applaud the authors for the clear amount of effort and thought that went into collecting and describing this dataset!

**Correctness:**

The dataset's construction is relatively sound given their goals, though I do have concerns related to the high disagreement rate, mentions above. The baselines they establish for their benchmark seem sound.

**Documentation:**

The dataset collection process is quite well documented in the supplementary material. A URL to the dataset was provided. I believe the authors sufficiently described to baseline approach to reproduce it.

**Ethics:**

I believe the authors' discussion of the ethical concerns regarding this dataset (in their supplementary material) is sufficient.

**Relation To Prior Work:**

To my knowledge, the authors have sufficiently placed their work in context with related work.

**Summary And Contributions:**

The authors release a relatively large-scale dataset and benchmark for fact verification that includes both unstructured and structured information. They also provide some baseline models, demonstrating performance over their new dataset. Overall I think this is a strong paper tackling an important problem. The authors claim that their proposed dataset is the first which considers both structured and unstructured evidence for verification, while explicitly requiring the retrieval of evidence. My primary concerns, discussed later in this review, are related to the ecological validity of the task.

---

> ### Author Response · Authors · 2021-07-12
> **Author response**
>
> We want to thank the reviewer for the feedback and comments on our paper. The following points are aimed to address the reviewer’s raised concerns and have been incorporated in our paper:
>
> *How does the task design in this dataset compare to real-world fact-checking problems? To what extent could a high-performing model trained on this dataset be useful for real-world fact-checking tasks? To what extent might it be problematic?*
>
> We added a paragraph to Section 5.3 to address this point. Incorporating information from both text and tables for fact-checking enables the verification of more complex claims than previous large-scale datasets, ultimately enhancing the practical relevance of automated fact-checking systems. However, FEVEROUS still simplifies real-world claims substantially, by controlling many variables of the claim generation process. For instance, it ignores the common strategy of biased evidence employed for generating misleading claims in the real world, also referred to as cherry picking, where facts which are true in isolation are being taken out of context, resulting in an overall false claim. As cherry picking is commonly done over numerical data we believe that considering both textual and structured data is a requirement for tackling this challenge.
>
> *The inter-rater agreement rate is quite low. The authors note that this rate is similar to that found in related datasets, and that the rate becomes very high when annotators consider the same evidence for a given claim. I'd like to have a better understanding of the implications of this disagreements. For instance, is it often the case that different evidence results in entirely supporting or refuting the exact same claim? Or is it more often that some evidence supports a claim, while other evidence simply doesn't provide enough information? Are there clusters of annotators who tend to agree or disagree with each other, or does it appear to simply be noise?*
>
> We added a paragraph regarding our analysis of verdict disagreements to section 3.2. We observe that in a third of two-way IAA disagreement cases an annotator selected NEI, which is disproportionately high considering NEI claims make up only 5\% of the dataset, indicating that the retrieval of evidence is a crucial part of the verification task. For the other two-thirds, when annotators selected opposite veracity labels we identified four sources of disagreement: (i) numerical claims that require counting a large number of cells, so small counting errors lead to opposing labels (ii) long claims that require many pieces of information to be verified and some part of a claim that was refuted had been overlooked (iii) not finding evidence that refutes/supports a claim due to relevant pages being difficult to find (e.g.\ when the article's title does not appear in the claim) (iv) accidental errors/noise, likely caused by the complexity of the task. Looking into the IAA between every annotator pair shows an overall consistent annotation quality with a standard deviation of 0.07 and a total of 10 annotators with an average IAA of below 0.60 and 8 with higher than 0.70.

---

> > ### Comment · Reviewer_yzsY · 2021-07-14
> > **thanks**
> >
> > My thanks to the authors for their response! I leave my score unchanged and remain positive about this paper.

---

### Official Review · Reviewer_wmbf · 2021-07-04
**Review of "FEVEROUS: Fact Extraction and VERification Over Unstructured and Structured information"**

**Rating:** 7
**Confidence:** 3

**Strengths:**

The use of the Grice’s maxims is commendable. I would encourage the authors, however, to verify the maxim of quantity, as it’s not only about sufficient information, but should also evaluate whether the evidence provides too much information. Furthermore, there is no actual reference back in 3.2 to show that the claims were of high quality.

The discussion section touches upon several important observations that shows possible avenues for future work.


**Weaknesses:**

In the Claim Generation section, 3.1.1, the authors mention that the annotators could highlight either four consecutive sentences or a table, which has at least 5 rows and at most 50 rows. Isn’t this a bit restrictive? How were all these thresholds chosen? At the moment, they seem fairly arbitrary. How were the restrictions implemented in the Claim using highlight only and beyond highlight - were there any checks implemented to make sure they don’t use paraphrases, or that they combine information from several rows in a table?

The link between the claim generation and claim verification is a bit ambiguous. In the claim generation, could the annotators create all three types of claims? Or only supported ones? Was the claim verification step performed by the same or different annotators? This detail comes too late in the paper and generates many unneeded questions.

The calibration section discusses approval rates for annotators - are the numbers provided in the “annotators” section still applicable or should we consider that only half were employed to create the dataset? There are unclear references to two-way and three-way IAA: by how many annotators were the claims verified? Did they make a binary decision for each reason, i.e., ungrammatical = true/false, typographical errors = true/false? Did annotators also have to agree on the state of the claim and the highlight? Overall, there are too little details regarding the entire annotation process, the reliability of the dataset and how this was computed. There is no discussion provided regarding the suitability of a kappa score of 0.65. What is still not clear, is how did the annotators reach agreement? How was the claim, final label, and the evidence chosen?

Authors provide details regarding data imbalance, but only for NEI claims. What about the other two types of claims? It would be nice to give some more details regarding the annotated dataset, especially in terms of overview, as a summary of the annotated dataset (this is again a bit late in the paper).

The results on classification should be compared with existing literature.


**Additional Feedback:**

Detailed comments are given in weaknesses section. The supplementary material does contain many of the issues that I raise. However, to improve the current manuscript, I strongly believe some details can be improved.

**Clarity:**

The paper is easy to read and follow. However, there are certain sections that could have an improved clarity, especially in terms of data annotation.

**Correctness:**

I would encourage the authors to compare their results with existing literature.

**Documentation:**

The dataset is publicly available.

**Ethics:**

No ethical concerns.

**Relation To Prior Work:**

The related work makes a nice overview of existing datasets for fact checking. The section, however, does not provide any details regarding the novelty of the approach presented in the current paper, which methods are extended or adapted from the literature. While Table 1 gives quite a nice overview of existing dataset and their characteristics, the Table is never described nor referenced. Also, not clear how these datasets were selected and how complete the comparison is.

**Summary And Contributions:**

The paper introduces a new dataset and benchmark called FEVEROUS, which consists of more than 87k claims. The claims are labeled as supporting, refuting and inconclusive, if there is support for the claim found in Wikipedia sentences and tables, if there is no support found in Wikipedia sentences and tables and, respectively, if there is not enough information to reach a verdict.

---

> ### Author Response · Authors · 2021-07-12
> **Author response**
>
> We thank the reviewer for the constructive feedback. The following addresses the points/questions and describes the adjustments we have made to our paper.
>
> *I would encourage the authors, however, to verify the maxim of quantity, as it’s not only about sufficient information, but should also evaluate whether the evidence provides too much information.*
>
> We measured the cases in which the evidence provides more information than required and added our insights into section 3.2: In 27\% of the claims the evidence of one annotator is a proper subset of another, indicating that in some cases evidence might provide more information than required, but on the whole, we believe this acceptable given the complexity of the task. Thus, the maxim of quantity largely holds for the evidence also in the other direction of only containing as much information as needed.
>
> *Furthermore, there is no actual reference back in 3.2 to show that the claims were of high quality. Were there any checks implemented to make sure they don’t use paraphrases, or that they combine information from several rows in a table?*
>
> While the quality of generated claims is very difficult to measure quantitatively, all claims produced were checked manually in the context of the verification step. The paper now also includes the proportion of claims that require a single sentence or cell as evidence, indicating to which extent claims adhere to the requirement of generating a claim not by simply paraphrasing a single sentence or cell.  Using two-way agreement, only 13\% and 7\% of non-refuted claims (as refuted claims only consist of evidence of the refuted part)  require a single sentence or cell as evidence respectively.
>
> *How were all these thresholds chosen?*
>
> These thresholds were put in place in order to filter out tables and sentences that had too little or too much information to generate claims with, respectively. Similar thresholds and restrictions were used in the creation of Tabfact that restricts its dataset to tables of 50 rows max too. This information has now been added to sec. 3.1.1.
>
> *In the claim generation, could the annotators create all three types of claims? Or only supported ones?*
>
> We have now added this information for each claim type in sec. 3.1.1. Only for the first claim we also specified the veracity of the generated claim, enforcing an equal balance between supported and refuted claims. For both Type II and III claims, annotators could freely decide to create either supported, refuted, or NEI claims, as long as they adhere to the claim requirements.
>  Was the claim verification step performed by the same or different annotators? This detail comes too late in the paper.
> Thanks, we added this information in sec. 3.1 now.
>
> *Are the numbers provided in the “annotators” section still applicable?*
>
> Yes, the numbers mentioned are applicable. We clarified this aspect in the paper (sec. 3.2) by specifying the total number of annotator candidates as well now.
>
> *by how many annotators were the claims verified?*
>
> The samples were selected randomly proportionally to the number of annotations by each annotator. This detail has now been added to sec. 3.2.
>
>
> *Did they make a binary decision for each reason?*
>
> We clarified the section in our paper, adding that annotators had to specify the reason(s) by selecting one or multiple predefined reasons and/or formulating an individual reason when reporting a claim.
>
> *What is still not clear, is how did the annotators reach agreement? How was the claim, final label, and the evidence chosen?*
>
> The majority of annotations were done by only one annotator per claim, and therefore there is no separate adjudication stage. Duplicate annotations (and hence disagreements) created as part of the FEVEROUS quality assurance are not considered for the dataset itself. We clarified this in sec. 3.2 now.
>
> *Authors provide details regarding data imbalance, but only for NEI claims. What about the other two types of claims? It would be nice to give some more details regarding the annotated dataset, especially in terms of overview, as a summary of the annotated dataset (this is again a bit late in the paper).*
>
> We added an additional row in Table 1 describing the distribution of veracity labels and further added in sec. 3 a summary of the annotated dataset based on the table.
>
> *The results on classification should be compared with existing literature.*
>
> Since related datasets either employ different metrics (e.g. FEVER does not consider evidence for evaluating NEI samples, see sec. 3.1.2) or are employed in a different setting (TabFact, Sem-tab-facts) the results of our model are not directly comparable to them.

---

### Decision · Program_Chairs · 2021-07-26

**Decision:**

Accept

**Comment:**

This paper introduces a dataset for fact-verification using both structured and unstructured data from WIkipedia.  Reviewers found the paper and dataset well motivated, documented, and novel.